Environmental influences on the Indo–Pacific octocoral Isis hippuris Linnaeus 1758 (Alcyonacea: Isididae): genetic fixation or phenotypic plasticity?

Rowley Sonia J. 1 2 srowley@hawaii.edu
Pochon Xavier 3 4
Watling Les 5 6
1 Department of Geology and Geophysics, University of Hawai’i at Mānoa , Honolulu, HI , USA
2 Department of Natural Sciences, Bernice Pauahi Bishop Museum , HI , USA
3 Coastal and Freshwater Group, Cawthron Institute , Nelson , New Zealand
4 Institute of Marine Science, University of Auckland , Auckland , New Zealand
5 Department of Biology, University of Hawai’i at Mānoa , Honolulu, HI , USA
6 Darling Marine Center, University of Maine , Walpole, ME , USA
Robinson Laura
Electronic publication date: 2015 Aug 20
Publication date: 2015
Volume: 3
Electronic Location ID: e1128
Received 2015 Apr 28; Accepted 2015 Jul 5
Copyright: © 2015 Rowley et al.
Copyright year: 2015
Copyright holder: Rowley et al.
License: This is an open access article distributed under the terms of the Creative Commons Attribution License, which permits unrestricted use, distribution, reproduction and adaptation in any medium and for any purpose provided that it is properly attributed. For attribution, the original author(s), title, publication source (PeerJ) and either DOI or URL of the article must be cited.
License URL: https://creativecommons.org/licenses/by/4.0/

Keywords: Wakatobi Marine National Park (WMNP), ITS2, Indonesia, Morphology, Isididae, Isis hippuris, Gorgonian coral

Funding: Victoria University Wellington Doctoral Research Scholarship Coral Reef Research Unit, University of Essex UK SJR was supported by a Victoria University Wellington Doctoral Research Scholarship and the Coral Reef Research Unit, with sponsorship from P Duxfield at Cameras Underwater© for camera equipment. The funders had no role in study design, data collection and analysis, decision to publish, or preparation of the manuscript.

==============================
As conspicuous modular components of benthic marine habitats, gorgonian (sea fan) octocorals have perplexed taxonomists for centuries through their shear diversity, particularly throughout the Indo–Pacific. Phenotypic incongruence within and between seemingly unitary lineages across contrasting environments can provide the raw material to investigate processes of disruptive selection. Two distinct phenotypes of the Isidid Isis hippuris Linnaeus, 1758 partition between differing reef environments: long-branched bushy colonies on degraded reefs, and short-branched multi/planar colonies on healthy reefs within the Wakatobi Marine National Park (WMNP), Indonesia. Multivariate analyses reveal phenotypic traits between morphotypes were likely integrated primarily at the colony level with increased polyp density and consistently smaller sclerite dimensions at the degraded site. Sediment load and turbidity, hence light availability, primarily influenced phenotypic differences between the two sites. This distinct morphological dissimilarity between the two sites is a reliable indicator of reef health; selection primarily acting on colony morphology, porosity through branching structure, as well as sclerite diversity and size. ITS2 sequence and predicted RNA secondary structure further revealed intraspecific variation between I. hippuris morphotypes relative to such environments (ΦST = 0.7683, P < 0.001). This evidence suggests—but does not confirm—that I. hippuris morphotypes within the WMNP are two separate species; however, to what extent and taxonomic assignment requires further investigation across its full geographic distribution. Incongruence between colonies present in the WMNP with tenuously described Isis alternatives (Isis reticulata Nutting, 1910, Isis minorbrachyblasta Zou, Huang & Wang, 1991), questions the validity of such assignments. Furthermore, phylogenetic analyses confirm early taxonomic suggestion that the characteristic jointed axis of the Isididae is in fact a convergent trait. Thus the polyphyletic nature of the Isididae lies in its type species I. hippuris, being unrelated to the rest of its family members.

Introduction

Reef biodiversity reflects that of its environment and geological history, those within the Indo–Pacific Coral Triangle some of the most diverse of all. Intense competition in such environments may lead to niche partitioning through resource acquisition, leading to ecological divergence. Such diversification may occur with or without extrinsic barriers to gene flow and is particularly marked in sessile modular organisms such as cnidarians, far from passive to their environment (Cossins et al., 2006; Prada, Schizas & Yoshioka, 2008; Prada & Hellberg, 2013) with respect to growth form and biochemical composition. However, delimitation between closely related species across steep environmental gradients on coral reefs may be confounded by phenotypic plasticity, homoplasy, cryptic or sibling taxa (Knowlton, 1993). It is, therefore, necessary to define environmentally driven divergent mechanisms on select phenotypic traits to accurately assess species biodiversity and endorse effective conservation management strategies (Ladner & Palumbi, 2012).

Gorgonian corals (Cnidaria: Anthozoa: Octocorallia) are conspicuous modular organisms, the greatest diversity occurring within the Indo–Pacific coral triangle yet remarkably ‘poorly known’ (Van Ofwegen, 2004). Intra- and inter-specific morphological variability in gorgonians is influenced by environmental factors such as light, temperature, sedimentation and flow rates (West, Harvell & Walls, 1993; West, 1997; Skoufas, 2006; Prada, Schizas & Yoshioka, 2008; Prada & Hellberg, 2013). However, little is known about the responses of gorgonian taxa to environmental parameters within the Coral Triangle. Distinct morphotypes of the isidid gorgonian Isis hippuris Linnaeus, 1758 exist between healthy (Ridge 1) and degraded (Sampela) reefs within the Wakatobi Marine National Park (WMNP), SE Sulawesi, Indonesia; short-branched multi/planar colonies, and long-branched bushy colonies, respectively (Fig. 1; Rowley, 2014; Rowley & Watling, in press). Whether such morphological differentiation is a consequence of a capacity to be plastic, plasticity as an adaptation (Hoogenboom, Connolly & Anthony, 2008), or has become genetically fixed leading to two species, is unclear.

Figure 1 Isis hippuris morphotypes and location map of the Wakatobi Marine National Park (WMNP), SE Sulawesi, Indonesia.

Isis hippuris morphotypes: (A) short branched predominantly planar or multiplanar colonies at the healthy site Ridge 1, and (B) long branched bushy colonies at the impacted site Sampela, with additional (C) collection localities within the WMNP. Sample number in brackets for molecular and asterisk for morphological analyses.

Isis hippuris within the WMNP may represent ‘robust’ canalisation where morphotypes are in fact two previously diverged species through disruptive (in sympatry) selection on traits between environments (Schluter, 2001). Alternatively, physiological developmental constraints may have become decanalised through an acute perturbation (e.g., Bergman & Siegal, 2003). In the first scenario, existence in low water velocity, high turbid environments typical of lagoon, semi-lagoon or sea-grass beds, as seen in Sampela, gave rise to an accumulation of pre- or postzygotic isolation between populations leading to separate adaptive fitness peaks representing ecological niches of long standing. Divergent morphotypes would therefore be robust to environmental change, maintaining native phenotypic traits. In the second scenario, cryptic variation more likely to be adaptive than random mutations, facilitate rapid mutation, as can acute perturbations (Flatt, 2005). Either case can be accelerated by pleiotropy, linkage disequilibrium or concerted evolution (Sánchez & Lasker, 2003) leading to population level genetic assimilation of a particular phenotype and thus provides a testable level of emergent trait integration (i.e., characters behaving as a unit). Moreover, phenotypic variation can be largely attributed to rapid evolutionary events (Eldredge & Gould, 1972; Simpson, 2013).

The phenomenon of species delimitation in closely related modular organisms can be investigated through models of integration (Magwene, 2001). Growth persists through the iterative addition of modules (e.g., polyps, branching properties), which may develop independently or in concert (by trait integration, see Magwene, 2001; Sánchez & Lasker, 2003; Sánchez et al., 2007) leading to differential integration in response to environmental perturbations. The co- and multi-variance of certain phenotypic traits may differ due to inextricably linked developmental (e.g., heterochrony; Sánchez, 2004) or functional integration leading to patterns of diversity through plasticity or divergent selection directly (extrinsic) or indirectly (intrinsic) on traits between populations or subpopulations (Schluter, 2001). By measuring five morphological traits in twenty-one Caribbean gorgonian species, Sánchez & Lasker (2003) revealed integration (coordinated character change) within both branching and polyp dynamics, yet the two were independent of each other. Furthermore, colony form and growth via branching were interconnected through the ratio of ‘mother’ branches to ‘daughter’ branches (Sánchez & Lasker, 2004). Whether this is replicated across all gorgonians i.e., from different regions and habitats, is unclear, however species-specific trait integration, particularly in response to environmental change, has been shown in other taxonomic groups (e.g., plants; Xu, Schooler & Van Klinken, 2012).

Isis hippuris may simply possess high capacity for plasticity, itself an adaptation facilitating considerable physiological tolerance to environmental heterogeneity, not uncommon in gorgonians (West, Harvell & Walls, 1993; West, 1997; Skoufas, 2006). Long-branched bushy, porous colonies reduce sediment settlement and maximise light capture through increased surface area and decreased self-shading in reduced light and water flow environments, as seen in the scleractinian Stylophora pistilata Esper, 1797 (Shaish, Abelson & Rinkevich, 2006). Whereas the densely packed short branches of planar colonies in high light and water flow, coupled with greater densities of small micro-skeletal elements (sclerites) provide mechanical strength (West, Harvell & Walls, 1993; Kim et al., 2004; but see Skoufas, 2006). Sclerites are key characters for species delineation within Octocorallia, those of the coenenchyme (soft tissue or ‘rind’) surface and polyps being most susceptible to environmental variation (Bayer & Stefani, 1987). Reduced polyp density with depth as a function of light in zooxanthellate taxa (West, Harvell & Walls, 1993; Kim et al., 2004; Prada, Schizas & Yoshioka, 2008; Prada & Hellberg, 2013) increases photosynthetic gain through surface area, yet polyp dimensions decouple integration by remaining independent of branching dynamics in Caribbean taxa (Sánchez & Lasker, 2003; Sánchez & Lasker, 2004; Sánchez et al., 2007). A broad trait assessment including genetic analyses would therefore provide further insight into the relationship between I. hippuris and its environment. Are such trait patterns commensurate with those described in other gorgonians, and thus fixed within the phylogenetic group (e.g., Sánchez & Lasker, 2003; Sánchez, 2004; Sánchez et al., 2007)?

A single representative of its genus, I. hippuris is taxonomically problematic (Watling et al., 2012), having a recognised plasticity (Wright & Studer, 1889; Simpson, 1906; Thomson & Simpson, 1909; Bayer & Stefani, 1987; Fabricius & Alderslade, 2001), which may obscure any possible species boundaries. In order to fully elucidate the nature of I. hippuris phenotypic variation among reef sites, a brief taxonomic and historical account is presented (see also Supplemental Information 1) with subsequent investigation into potential adherence to previously documented lower taxonomic assignments.

The family Isididae Lamouroux, 1812 [nom. correct. Kükenthal, 1915 (pro Isidae Lamouroux, 1812)], currently placed within the sub-Order Calcaxonia, is characterised by a unique axis of alternating calcareous internodes and proteinaceous (gorgonin) nodes giving a bamboo appearance. Calcareous internodes can be hollow or solid and are not sclerobastic (i.e., consisting of fused sclerites sensu the sub-group Scleraxonia). The Isididae was further subdivided into four currently accepted subfamilies (see Alderslade, 1998; Watling et al., 2012) based primarily on polyp retractability and sclerite composition and arrangement. Pertinent to this study, the sub-family Isidinae Lamouroux, 1812 (sensu Studer, 1887) is distinguished by small, warty and usually irregular sclerite forms and contains the genera Isis Linnaeus, 1758 and Chelidonisis Studer, 1890. Within the genus Isis 20 species have been assigned over the years with only I. hippuris Linnaeus, 1758, currently recognized as valid. Isis hippuris Linnaeus, 1758 is the type species for the Isididae, Isidinae and genus Isis and its validity widely accepted (Bayer & Stefani, 1987; Fabricius & Alderslade, 2001). Isis reticulata (Nutting, 1910) and I. minorbrachyblasta (Zou, Huang & Wang, 1991) have occasional reference but with taxonomic misgivings (see Bayer & Stefani, 1987; but also Mai-Bao-Thu & Domantay, 1971). Diagnostic traits are, therefore, summarised here (see also Supplemental Information 1) to compare with those found within the WMNP.

Isis hippuris colonies are arborescent, planar or bushy with varying branch length and diameter, which can be swollen at the tip. Polyps are distributed all around the branches. A diversity of sclerites within the coenenchyme includes small warty clubs and spindles, asymmetric capstans, doubleheads and occasional crosses (e.g., Bayer & Stefani, 1987). Colonies of I. reticulata are slender, planar, with long slim branches, and asymmetrically distributed polyps. Sclerites are generally smaller, and their warts, when present, are arranged symmetrically (e.g., Nutting, 1910). Isis minorbrachyblasta colonies are bushy with densely aggregated short, fine branches bearing equally distributed polyps. Sclerites are also variable in form and articulation. In sum, diagnostic phenotypic traits tend mostly to place I. minorbrachyblasta as an intermediate between I. hippuris and I. reticulata.

Isis within the Wakatobi—Isis morphotypes found within the WMNP bear only partial adherence to those previously described (see Supplemental Information 1) at the colony level. The long-branched bushy colonies on degraded reefs, and short-branched multi/planar colonies on healthy reefs may represent I. reticulata and I. hippuris respectively, or simply the widely accepted plasticity of the latter (Wright & Studer, 1889; Simpson, 1906; Thomson & Simpson, 1909; Bayer & Stefani, 1987; Fabricius & Alderslade, 2001), likely through an integration effect (Magwene, 2001; Sánchez & Lasker, 2003).

Clearly Isis taxonomy is in a state of confusion, compromising field identification and subsequent conservation efforts due to difficulties in species assignment. A thorough examination of Isis specimens throughout its distribution is both necessary and underway, yet, outside of the scope of this study. Therefore, given the tenuous nature of previously described Isis species with the exception of I. hippuris, from here on in all specimens will remain assigned to I. hippuris, unless specified otherwise.

Here an assessment of morphotypes found within the WMNP relative to reef health is presented, with brief comparisons to those previously described. Firstly we ask: are populations of I. hippuris morphotypes phenotypically and genetically subdivided due to contrasting reef environments within the WMNP, Indonesia? Secondly, do the I. hippuris morphotypes represent previously described species, or a single species with highly variant, integrated phenotypic traits? Therefore, this study aims to: (1) investigate morphological variability in the zooxanthellate gorgonian I. hippuris between contrasting coral reef environments within the WMNP, SE Sulawesi, Indonesia; (2) identify patterns of genetic variability relative to such morphotypes using population genetics and predicted RNA secondary structure of the nuclear ribosomal Internal Transcribed Spacer region 2 (ITS2), (3) to subsequently infer mechanisms of speciation or phenotypic plasticity as a consequence of environmental change, and (4) investigate the currently assigned phylogenetic position of I. hippuris within the Octocorallia using the ITS2 region.

Materials and Methods

Study area

The Wakatobi Marine National Park (WMNP) is a remote archipelago of ca. 13,900 km2 in S.E. Sulawesi, Indonesia. The epicentre of the Coral Triangle and Indonesia’s second largest marine park, the WMNP comprises ca. 600 km2 of the most biodiverse coral reefs on earth. Such marine biodiversity sustains >100,000 people with considerable human population expansion and consequential marine resource dependence and destructive commercial fisheries (Clifton, 2013). Coral reefs within the Wakatobi range from low current, high turbidity lagoons to highly exposed sites with strong water currents and high nutrient deep-water upwellings. Therefore, strong environmental gradients of natural and anthropogenic disturbance exist across reefs within the Wakatobi, providing a novel natural laboratory for studies of environmentally induced change on reef components.

Research was conducted during July and August 2010, between two sites spanning 5 km of anthropogenic and natural environmental gradients (Fig. 1C). Ridge 1 (healthy) is an exposed reef ridge with high nutrient upwellings and water currents; Sampela (impacted) is a semi-lagoonal reef with low water flow and high turbidity. Sampela is situated ca. 400 m from a Bajo (sea gypsy) village of ca. 1,600 people and is subject to continuous marine resource exploitation and community waste disposal. Environmental variables likely to influence the distribution of morphotypes for the two study sites (summarized in Table 1) were selected for further analyses. Turbidity (NTU, expressed as inverse values), and chlorophyll-a (µg L−1) were measured using an RBR® XR-420 CTD data logger; a General Oceanics® flow meter was used to measure water flow velocity. Temperature (°C) and light (measured as lux and presented as Kd(PAR)) were measured using HOBO® data loggers. All loggers were placed in the upright position at each test site at 3–5 m depth, recording every minute for up to 24 h cycles for at least the study period (temperature was measured annually recording every 15 min, 2007–2011). Suspended sedimentation rates were assessed using four standard 1.0 litre sediment traps (English, Wilkinson & Baker, 1997) deployed at each site for a 10-day period (2006–2011, June–August, and October–November in 2008). Sediment and water were filtered (Whatman 0.2 µm pore size), dried at 60 °C and weighed with rates expressed as g dry weight day−1. Sediment grain size was estimated using Retsch Technology® test sieves, with logarithmically converted diameters expressed as phi (Φ) and classified using the Wentworth scale (Wentworth, 1922). Environmental variables, with the exception of latitude and longitude, were edited visually with significant outliers removed, and entered into statistical models as raw values.

Table 1 Table of study sites environmental characteristics.

Environmental characteristics of the two study sites in the WMNP, Indonesia. All values expressed as mean (±SE) with the exception of diurnal temperature range (°C), light (Kd(PAR)) and sediment grain size (Φ).

Parameter recorded	Mean value ± SE or range	
Site	Sampela	Ridge 1	
Latitude (S)	005°29′01″	005°26′57″	
Longitude (E)	123°45′08″	123°45′38″	
Temperature (°C min–max)	25.61–29.36	24.06–28.07	
Light (Kd(PAR) min–max)	0.31–3.14	0.1–1.56	
Flow (cm/s)	5.02 ± 2.18	30.54 ± 2.61	
Chlorophyll-a (μg 1−1)	0.3 ± 0.01	0.35 ± 0.03	
Turbidity (NTU)	4.38 ± 1.80	0.17 ± 0.33	
Sedimentation (g d−1, n = 12)	3.28 ± 0.26	1.16 ± 0.07	
Sediment grain size (Φ, n = 12)	5 [31.25–62.5 μm]	1 [0.5–1 mm]	

Sample collection

Isis hippuris colonies were sampled from the healthy, site Ridge 1 (n = 24), and the anthropogenically-impacted site, Sampela (n = 24; total n = 48), where the two distinct morphotypes at densities of 18 and 6 colonies per 10 m−2, respectively, were previously documented (Rowley, 2014; Rowley & Watling, in press; Figs. 1A and 1B). A further twelve clippings were taken from five additional sites (Blue Bowl n = 2; Pak Kasim’s n = 2; Buoy 3 n = 4; Kaledupa n = 2; Sea Grass beds n = 2; Fig. 1C) to investigate and compare genetic differences between colonies within the study area. Sample numbers for molecular analyses were low due to financial constraints, yet provide valuable insights for further study. All colonies were randomly selected within 2–5 m depth and a minimum of 10 m distance apart to avoid sampling asexual clone mates. Each colony was subject to in situ scaled digital photography using a Canon IXUS 900Ti, WP-DC7 u/w housing and INON UWL-105 AD × 0.51 lens, with duplicate samples preserved in 95% EtOH and Guanidinium solution for morphological and molecular analyses respectively. Colonies were photographed both parallel and overhead for planar and bushy colonies as appropriate with a ruler for scale. The State Ministry of Research and Technology (RISTEK: No. Surat. Izin: 014/SIP/FRP/SM/VI/2010) granted research permits to Prof DJ Smith, under whose auspices this work was conducted.

Data analyses

Morphological measurements

Population comparisons of morphological traits were conducted on 24 I. hippuris colonies from both Ridge 1 and Sampela (total 48 colonies). A total of 57 morphological traits were quantified and divided into 32 macro- and 25 micro-morphological traits (Fig. 2; Table S2). Due to the variable nature of I. hippuris colonies (planar, multiplanar or bushy), particularly between sites, whole colony height (H), mean width (W) taken equidistant apart, and colony spread (CS) were measured with CS as the mean of two measurements taken above the colony. Branch tips (T), mid main branch (M) and base (B) width were also recorded but limited access to the latter meant data were omitted from further analyses (n = 13 and 5 for Ridge 1 and Sampela respectively). Colony sub-sections of ∼20 cm in height were selected for further comparable macro-morphometric analyses: sub height (sH), mean width (sW), and projected sub-colony area (PA) estimated by sH × sW. PA was then used to calculate sub-colony porosity (Po) as a ratio of PA and the projected branch area (PBA) (total branch length multiplied by the mean branch thickness; see below). Branch articulation was assessed using a hierarchical generation ordering system (Lasker et al., 2003; Sánchez & Lasker, 2003; Sánchez, McFadden & France, 2003), where each branch was ascribed as either a ‘mother’ branch or ‘daughter’ branch, the latter emerging from the former. As the colony develops, daughter branches may also become mother branches (e.g., second generation mother branch; see Fig. 2E) quantified as follows: mother branch length (sML), mean mother branch width (sMW), daughter branch length (sDL), mean daughter branch width (sDW), total branch number (sTB#), total branch length (sTBL), and mean branch width (MBW). Branch surface area was calculated on the geometric approximation of a cylinder from branch length and mean width as the radius, with subsequent polyp density (PD) per cm2. Twenty random measurements were made of both polyp diameter ((pD) mean of 2 measurements see Fig. 2D) and inter-polyp distance (ID). All polyp, branch cross-section and canal (C#/Cd; Cd see Fig. 2C) quantification were visualised under an Olympus SZX16® stereomicroscope at 10×magnification with 0.5× objective.

Figure 2 Isis hippuris morphological trait measurements plate.

Isis hippuris morphological trait measurements of the (A) colony; (B, C, D) canal and polyp dynamics; (E) sub-colony (branching) dynamics; (F) sclerites site/morphotype comparisons of i and ii spindles, iii and iv capstan 7-radiates, v–vi and vii–viii clubs from Ridge 1 and Sampela respectively. All abbreviations are described in text.

Isis hippuris has a considerable diversity of sclerite form (Simpson, 1906; Bayer & Stefani, 1987; Fabricius & Alderslade, 2001; see Figs. S1 and S3). For consistency, only those represented in all test colonies were selected for quantitative analyses. Length and mean width of three measurements were made on 20 randomly selected sclerites per sclerite type; surface clubs (CL1/2, CW1/2); and sub-surface capstans (7-radiates: CaL/W) and spindles (SL/W) (Figs. 2Fi–viii). Additional sclerite diversity is shown in Fig. S3. Sclerites were removed by dissolving the surrounding tissue in 5% sodium hypochlorite solution and visualized using optical microscopy (Olympus BX51®) and scanning electron microscopy (SEM), which was performed on a Hitachi S-800 SEM at the University of Hawai’i at Mānoa, USA. All micro-morphological measurements and sclerite preparation were taken 2 cm below the branch tip to avoid underdeveloped traits due to sub-apical branch growth (Lasker et al., 2003) and photographed using an Olympus 3.3MPX™ camera and Rincon software (ImagingPlanet®). All macro- and micro-morphological characteristics were measured using ImageJ64 (Abràmoff, Magalhaes & Ram, 2004).

Phenotypic traits were analysed using routines within the PRIMER-E v6.1.12 statistical package (Clarke & Gorley, 2006), with PERMANOVA+ v1.02 extension (Anderson, 2001). Specifically, character traits (untransformed) were simultaneously correlated in a Draftsman plot to eliminate uninformative traits (those showing no significant difference between test factors and factor levels; P > 0.95) and to establish appropriate transformation for downstream analyses (Clarke & Ainsworth, 1993). Informative phenotypic trait data were subsequently standardized and a ‘zero-adjusted’ Bray-Curtis similarity matrix (Clarke, Somerfield & Chapman, 2006) constructed for tests of morphological divergence between the two sites; Ridge 1 and Sampela. A single-factor model with 9,999 permutations (PERMANOVA; Anderson, 2001) was performed and further visualised utilizing constrained canonical analysis of principal coordinates (CAP; Anderson & Willis, 2003). Informative traits contributing most to the dissimilarities between sites, thus specific morphotypes were investigated using similarity percentages (SIMPER; Clarke, 1993) and displayed as a vector overlay on the CAP ordination.

The relationship between I. hippuris morphotypes and their environment was investigated using nonparametric multivariate regression (McArdle & Anderson, 2001) with the DISTLM forward routine (Anderson, 2003). Based on a Euclidean distance matrix, all raw environmental variable data (Table 1) were normalised and significance tested using 9,999 permutations (Anderson, 2001).

Molecular analyses

Genomic DNA of I. hippuris were extracted from 28 colonies, 8 from each of the two test sites (n = 16) and 12 from additional site populations for area and morphotype comparison as described above. Approximately 2–3 mm of fresh soft tissue was immediately cut and stored in 400 µl Guanidinium lysis buffer (4 M guanidinium isothiocyanate, 0.05 M Tris pH 7.6, 0.01 M EDTA, 0.07 M Sarkosyl, β-mercaptoethanol 1% v/v) (Pochon, Pawlowski & Zaninetti, 2001) for 14 days at room temperature during transit from the field, with subsequent storage at 4 °C. Preserved samples were incubated at 72 °C for 20 min, vortexed prior, during and after incubation, then centrifuged at 16,000 g for 5 min. The resulting DNA-containing supernatant was precipitated with an equal volume of 100% isopropanol, vortexed and stored over night at −20 °C. DNA was pelleted via centrifugation at 16,000 g for 15 min, washed with 70% EtOH, centrifuged for 10 min, dried and resuspended in 0.1 M Tris Buffer pH 8. The DNA solution was placed on ice for 1 h with frequent vortexing and stored at −20 °C. DNA was visualized on 1% agarose gel. PCR amplifications of the ITS2 rDNA marker were conducted using the primers itsD (forward; 5’-GTGAATTGCAGAACTCCGTG-3’) and ITS2Rev2 (reverse; 5’-CCTCCGCTTACTTATATGCTT-3’) (Pochon & Gates, 2010). Total PCR volume was 50 µl constituting: 5.0 µL of 10× PCR Buffer (Bioline Incl., London, UK), 2.0 µL of MgCl2 (2 mM), 1.0 µL of each primer (10 mM), 1 µL (2.5 mM of each dATP, dCTP, dGTP, and dTTP), 0.2 µL of Hotstart Immolase Taq polymerase (Bioline Incl., London, UK), 1.0 µL of DNA, and 39 µL of sterile water. Touchdown amplification was conducted as follows: denaturation at 95 °C for 10 min, 25 cycles at 94 °C then 35 s at 65 °C (reduction in annealing temperature of 0.5 °C per cycle), and 2 min at 72 °C. A further 14 cycles of 30 s at 94 °C, 35 s at 52 °C, 2 min at 72 °C, and a final 10 min extension at 72 °C. All amplicons were purified using the QIAquick™ PCR Purification Kit (Qiagen, Hilden, Germany), and separated by cloning for haplotype verification. Purified products were ligated into the pGEM®-T Easy vector™ (Promega, Madison, Wisconsin, USA), transformed into α-Select Gold Efficiency™ competent cells (Bioline), with subsequent positive inserts verified by PCR using plasmid specific primers (M13). Positive inserts (8-12 per library) were purified with an ExoSAP-IT kit, sequenced in both directions using the ABI Prism Big Dye™ Terminator Cycle Sequencing Ready Reaction Kit and run on an ABI 3100 Genetic Analyzer (Perkin-Elmer Applied Biosystems, Foster City, California, USA) at the University of Hawai’i at Mānoa, USA.

ITS2 clone libraries from 28 individuals were aligned using ClustalW2 (Thompson, Gibson & Higgins, 2002) and manually edited in Geneious Pro v.5.6.2 (Biomatters Ltd., NZ). A selection criterion of identical sequences from two or more clone libraries was established to minimize the effect of intragenomic variation and/or PCR artefacts on further analyses. On average 4–6 host clones were recovered per library due to simultaneous recovery of both host and endosymbionts.

Estimates of genetic differentiation relative to morphotype were investigated via an analysis of molecular variance (AMOVA) with pairwise population comparisons (ΦST) between sites using ARLEQUIN v.3.5 (Excoffier & Lischer, 2010). Haplotype (hd), nucleotide diversity (π) and substitution rate (Jukes-Cantor (JC)) were calculated with DNAsp v.5.0 (Librado & Rozas, 2009). A parsimony haplotype network with a 95% confidence level and gaps treated as a fifth state was constructed using Network v.4.6.1.1 on sample sequences only.

ITS2 predicted RNA secondary structure

ITS2 RNA secondary structures were predicted to further investigate haplotype differences specifically between Ridge 1 and Sampela at a more conserved level. Alcyonium digitatum Linnaeus, 1758 (Genbank Acc. #AF262347; McFadden et al., 2001) was used as a template for conserved motif identification with subsequent constraints implemented into MFOLD (Zuker, 2003) using default parameters. RNA was folded at 37 °C and structures with the highest negative free energy values, thus stability, were selected, manually edited in 4SALE (Seibel et al., 2006; Seibel et al., 2008) and visually annotated in VARNA (Darty, Denise & Ponty, 2009).

Phylogenetic reconstructions between I. hippuris haplotypes and twenty octocoral ITS2 outgroups obtained from GenBank (see Table S4) were conducted using the plugins PHYML 2.1.0 (Guindon & Gascuel, 2003) and MrBayes v3.2.2 (Huelsenbeck & Ronquist, 2001) within Geneious. Indels (insertion and deletion mutations) were considered phylogenetically informative and treated as separate characters using the ‘simple indel coding’ gap method (Simmons & Ochoterena, 2000) in GapCoder v.1.0 (Young & Healy, 2003). Maximum Likelihood (ML) phylogeny was conducted using the best-fit model (Jukes-Cantor (JC)) of nucleotide substitution as selected in jModelTest 2 (Darriba et al., 2012) through Akaike Information Criterion (AIC). Bayesian Inference (BI) phylogeny was made with a JC69 substitution model. Analyses were initiated from a random tree of four chains with two runs of Metropolis-coupled Markov Chain Monte Carlo (MCMCMC), including 1,100,000 generations and subsampling every 10 generations. Chains converged within 0.2 generations in all cases with a burn-in of 100,000. Phylogenetic trees were rooted with Paragorgia kaupeka Sánchez, 2004 and node support values set at ≥70% for both ML and BI.

Results

Morphometrics

Of the 57 morphological traits measured (59,328 individual measurements), 32 were selected for further analyses (41,265 individual measurements, see Table S2). Based on these traits, PERMANOVA revealed significant differences between morphotypes across the two study sites Sampela and Ridge 1 (pseudo-F = 14.489; P < 0.0001), further corroborated by the CAP analysis (δ2 = 0.995, P =   < 0.0001; Fig. 3) with 89% variance (% var.) as the total variance explained by the first m PCO axes. Prominent morphological traits contributing most to dissimilarities between sites were primarily at the colony level (TW, W, H, PBA, Po, and sTB#) with the exception of a higher polyp density (PD) at Sampela (Fig. 3). From both Fig. 3 and Table S2 it is clear that larger colonies present at Sampela have a reduction in branch density, yet still exhibit overall larger colony size and spread (PBA, sTB#, Po, and H, W, TW respectively). Branches, including the branch tips, were also consistently longer and thicker, at Sampela; however, polyp parameters were relatively invariable despite significantly high polyp density. It is noteworthy that all sclerite trait measurements were consistently smaller at Sampela (see Table S2 and Fig. S3), particularly capstans (see Figs. S3F.iii, S3F.iv and Fig. 2 for variability), which are variable throughout Isis hippuris distribution (Simpson, 1906; Bayer & Stefani, 1987; Fabricius & Alderslade, 2001). Irrespective of pre-treatment, the magnitude of differences between sclerite measurements were not that of the colony level. Nevertheless, separation and re-analyses under the same models for macro (e.g., colony and sub-colony: pseudo-F = 15.255; P < 0.0001; CAP δ2 = 0.976, P =   < 0.0001, 91% var.) and micro (e.g., sclerite: pseudo-F = 11.727; P < 0.001; CAP δ2 = 0.996, P =   < 0.0001, 85% var.) measurements did not significantly alter results from the full model, demonstrating a lack of redundancy in selected character traits.

Figure 3 Constrained ordinations (CAP) of Isis hippuris character traits.

Constrained ordinations (CAP) of Isis hippuris character traits between Ridge 1 [◆] and Sampela [▴]. Vector abbreviations as described in text.

Results from the distance-based nonparametric regression (DISTLM forward) revealed that turbidity and sediment load explained 27.31% (pseudo-F = 5.100; P < 0.001) of morphotype differences between the two sites.

ITS2 Sequence diversity

From the 28 Isis samples 120 clones were recovered: Ridge (34), Sampela (29), Sea Grass (9), Kaledupa (12), Buoy 3 (18), Pak Kasim’s (10), and Blue Bowl (8); GenBank accessions KP265675–265702. ITS2 sequences revealed five haplotypes: 1–3 per sample population with up to 8 substitutions (see Table S5 and Fig. 4). In keeping with morphological traits, colonies found at Sampela were significantly different from all other sample sites (Table 2 and Fig. 4). Haplotype diversity was greatest across Hoga Island with overall haplotype (hd) and nucleotide diversity (π) measured as 0.780 and 0.0197 respectively with just (JC) 0.0313 substitutions per site. Population division was strongly inferred by all AMOVA models (Table 2) and haplotype network analysis, the latter showing no evidence of reticulation through homoplasy (Fig. 4). Curiously, the single haplotype present in the sea-grass beds (D) shared no nucleotide differences with Sampela (A) despite its relatively close proximity, however little can be determined without greater sampling effort. Pairwise ΦST estimates of ITS2 sequences from Sampela ranged from 1.000 (Sampela vs. Sea Grass, Kaledupa, Pak Kasim’s, Blue Bowl, and Buoy 3) to 0.843 (Sampela vs. Ridge 1; P < 0.0001 in all cases), and from 0.467 (Ridge 1 vs. Pak Kasim’s; P < 0.05) to no structure (Ridge 1 vs. Blue Bowl and Buoy 3) at Ridge 1. Note that such values, in particular the P-values, should be treated with caution. Low sample sizes reduce fine-scale structure detection. Thus more data would likely provide greater insight into the level of haplotype and nucleotide diversity observed across sites and also increase taxonomic certainty.

Figure 4 Isis haplotype network and ITS2 RNA predicted secondary structure.

Isis haplotype network with corresponding ITS2 RNA predicted secondary structure relative to haplotype ((A–E), see also Fig. 5) and enthalpy values according to MFOLD. Roman numerals (I–IV) represent helices; red and black arrows indicate point mutations and loop differences, respectively. Coloured bases according to transitions (red), transversions (yellow) and gaps (lilac), S5. Haplotype circle diameters are proportional to identical clone sequences.

Table 2 AMOVA of Isis hippuris genetic structure between study sites.

AMOVA of genetic structure between sites within the WMNP from both cloned and sample sequences. R1 denotes Ridge 1; S denotes Sampela.

Source of variation	df	SS	Variance component	Variance %	Φ ST	
7 Populations: Clones						
Among populations	6	122.248	Va = 1.237	80.97	0.80974*	
Within populations	113	32.844	Vb = 0.291	19.03		
Total	119	155.092	1.528			
7 Populations: Samples						
Among populations	6	33.768	Va = 1.401	76.83	0.76831*	
Within populations	21	8.875	Vb = 0.423	23.17		
Total	27	42.643	1.824			
2 Populations (R1 & S): Samples						
Among populations	1	17.688	Va = 2.161	84.32	0.84321*	
Within populations	14	5.625	Vb = 0.402	15.68		
Total	15	23.312	2.563			
Notes.

* P < 0.001 significant.

ITS2 predicted RNA secondary structure analyses revealed minimal variation between haplotypes with the exception of Sampela (Fig. 4) providing greater confidence in phenotypic trait differences. Clones were collapsed into haplotypes per sample for phylogenetic analyses. Phylogenetic topologies using ML and BI algorithms were very similar including all outgroups and unambiguously identical with regard to WMNP haplotypes (Fig. 5). Branch support was typically stronger with BI particularly regarding outgroup species where recognised taxonomic suborders and groups were distinct. Phylogenetic uncertainty leading to the addition of multiple outgroups, confirmed I. hippuris sequences from the WMNP were not grouped with morphologically described sister taxa within the Isididae (highlighted red, Fig. 5). Reducing the outgroup number did not alter the integrity of the phylogenetic signal, in fact irrespective of model or selected root Alcyonium digitatum consistently positioned directly above Isis haplotypes.

Figure 5 Phylogram of Isis haplotypes and Octocoral out groups.

Phylogram based on Maximum Likelihood (ML) analyses of the ITS2 region from twenty Octocoral taxa in GenBank and Isis haplotypes within the WMNP. Branch numbers represent ML bootstrap support and BI posterior probabilities, respectively, with low values expressed as a hyphen (–) ≤70% and asterisk (*) indicative of differences from MrBayes phylogenetic inference. Letters Sc, Scleraxonia; Ca, Calcaxonia, Ho, Holaxonia, Al, Alcyoniina, and A–E represent Isis haplotypes as depicted in Fig. 4.

Discussion

Isis hippuris morphotypes were clearly defined both morphologically and genetically between the two sites Ridge 1 and Sampela (ΦST = 0.7683, P < 0.001; Table 2). Even with a reduced ITS2 sample size, molecular differences were consistent with morphometric results indicating that divergence has or is taking place, the nature of which is unclear. Multivariate trait integration at the colony level (including branching parameters), polyp density and sclerite size define significant differences between morphotypes indicative of trait interdependency, yet polyp dimension and canal width appear canalised (genetically fixed). Nevertheless, inherent phenotypic plasticity and/or disruptive selection may enhance the success of two phenotypes particularly between contrasting environments. Trophic-level interaction through differential light and nutrient exposure may drive such phenotypic differences, further reinforced by population structure due to asexual fragmentation and external brooding (Rowley, 2014). Taxonomic assignment may be tenuous, however, considering the partial adherence of morphotypes to previously described species within the Isis genus in addition to polyphyly within the Isididae.

Of the 48 colonies (28 used for genetic analyses) studied here (and 1,094 ecologically surveyed in Rowley, 2014; Rowley & Watling, in press) it cannot be said with confidence that I. hippuris morphotypes at either site within the WMNP adhere to the descriptions as outlined for Isis reticulata (Nutting, 1910; Kükenthal, 1919; Kükenthal, 1924; Stiasny, 1940; Mai-Bao-Thu & Domantay, 1971) or Isis minorbrachyblasta (Zou, Huang & Wang, 1991). The various morphotypes may simply represent phenotypic response(s) to water depth in those previously described. Isis hippuris contrasts with I. reticulata on the basis of short thick branches in the former, and long thin branches with smaller, coarsely articulated sclerites in the latter. Isis minorbrachyblasta has bushy colonies with short densely packed branches, but considering both the lack of sampling for this taxon, documented panmixia and noted phenotypic plasticity of I. hippuris (e.g., Simpson, 1906; Thomson & Simpson, 1909; Bayer & Stefani, 1987), this latter taxonomic assignment is treated with extreme caution and may simply be an intermediate form. The two morphotypes of I. hippuris presented here have but partial adherence to those previously described (summarised in Supplemental Information 1). The short-branched predominantly planar colonies at Ridge 1 are more akin to I. hippuris whereas the more open, long-branched colonies at Sampela resemble I. reticulata but with thick branches as opposed to thin. Swollen branch tips characteristic of I. hippuris were observed in both morphotypes, and is not a reliable trait. Such swollen branch tips were more prevalent at Sampela. This was the only site where external brooding was observed at the time of the study (Rowley, 2014), and therefore, swollen branch tips may pertain to the presence of eggs within the polyps.

Isis hippuris phenotypic variability

Measuring a broad range of phenotypic traits between I. hippuris morphotypes highlights trait integration, canalisation and thus those traits acted on by selection which may differ from those previously described for other gorgonian taxa (e.g. Sánchez & Lasker, 2003; Sánchez, McFadden & France, 2003; Sánchez et al., 2007; Sánchez, 2004; Dueñas & Sánchez, 2009). Here clear patterns of colony (therefore branching), integration coupled with sclerite-level traits and polyp density are consistent between the two morphotypes. Specifically, branching dynamics and colony size (colony porosity as a function of total branch number and size (projected branch area)) appear to have a negative association with sclerite size. Whether these traits are negatively associated as emergent properties or of longstanding would necessitate further investigation using reciprocal transplant experimentation and population coalescence (Prada, Schizas & Yoshioka, 2008; Prada & Hellberg, 2013). In either case, differential light attenuation and nutrient components between the two sites are not unnatural phenomenon, which may or may not be exacerbated by reef resource dependent anthropogenic influence from Sampela.

Colony surface area and metabolism are intrinsically linked whereby a cascade effect of concomitant variations in branching, polyp, canal and sclerite dynamics would be expected. However, disintegration or canalisation (fixation) was evident in both polyp and canal dimensions consistent with previous work (Sánchez, 2004). Responses to variations in water quality, thus heterotrophic feeding capacity, are incurred through polyp density as opposed to size, yet both canal number and dimensions remained unchanged in both morphotypes. The exact function of stem canals is unclear (Cadena et al., 2010), although suggested to circulate and exchange water and nutrients throughout the coral colony (Ellis & Solander, 1786). Canalisation/fixation at this level further suggests that photosynthetic gain with nutrient translocation at the cellular level between endosymbionts and host, are likely the primary trophic resource. Optimal allocation theory posits an increase in the uptake of resource(s) that are most limiting growth (Weiner, 2004). Moreover, the same genotype can show resource allocation plasticity (Sebens, 1997) in alternate environments consistent with the ‘partitioning’ hypothesis (Weiner, 2004). Plasticity as a response is an emergent property of divergence (Pigliucci, 2005). Therefore, to further elucidate energy allocation patterns between morphotypes, physiological tests coupled with morphological and genetic analyses on reciprocal transplants between reefs would establish phenotypic trait plasticity—thus a capacity for plasticity—or ecological divergence through disruptive selection (Schluter, 2001) in I. hippuris.

Sclerite composition can vary with light intensity and/or water motion (West, Harvell & Walls, 1993; Kim et al., 2004; Skoufas, 2006; Clavico et al., 2007; Prada, Schizas & Yoshioka, 2008; Prada & Hellberg, 2013). The presence of numerous small, articulate interlocking sclerites could provide additional structural support for larger colonies found at Sampela, which lack the close branching structure present at Ridge 1. Smaller sclerites may mitigate mechanical constraints on the axis of increased colony size and bushy morphology through long thick branches, and provide greater soft tissue support as surface area increases (Clavico et al., 2007). Small clubs increase both flexion and torsion capacity in less exposed areas of Eunicella singularis Esper, 1791 whereas larger spindles were prevalent in the exposed peripheral branches. Nonetheless, Eunicella cavolinii Koch, 1887 showed no selective difference between the two (Skoufas, 2006). A decrease in sclerite size with increased density in shallow conspecifics has been shown (e.g. Prada, Schizas & Yoshioka, 2008), typically due to increased water flow (West, Harvell & Walls, 1993; West, 1997; Kim et al., 2004). Here, regardless of both morphotypes containing high densities of small sclerites, the consistency in small size at Sampela coupled with thicker longer branches and higher polyp density likely increases photosynthetic gain through greater surface area, as well as heterotrophic feeding. The lack of variability in canal size or number, as well as polyp dimensions represents canalisation, and further suggests that photosynthetic gain from dinoflagellate endosymbionts is the primary resource for I. hippuris.

Sclerites are key characters for species identification yet susceptible to environmental perturbation and selection. All sclerites were consistently smaller in colonies from Sampela. The overall dimensions between the two morphotypes from both sites were within the range of those described for I. hippuris with regards clubs and radiates from both I. reticulata and I. minorbrachyblasta. Sclerite differences between morphotypes compared to those published appeared inconclusive with notable overlap. For example, bent spindles characteristic of I. reticulata were present in colonies from Ridge 1, themselves bearing closer resemblance to I. hippuris. Interestingly, sclerite diversity was greater at Sampela with closer resemblance to I. reticulata, particularly considering sclerite asymmetry and crosses. No sclerites were found within the polyps or tentacles in either morphotype in this study, unlike in I. reticulata. However, the small rods (0.07 × 0.01 mm) of Bayer & Stefani (1987) were present, but their precise location within I. hippuris soft tissue could not be determined.

Enhanced fitness through an individual’s (genotype) capacity to respond to environmental heterogeneity—specific morphotypes predominating in certain habitats—maximises survival through resource acquisition and minimises metabolic costs. Most corals are polymorphic under varying environmental conditions (West, Harvell & Walls, 1993), with differential phenotypic expression of a genotype as a consequence of astogeny (colony development), itself genetically and/or environmentally mediated (Sánchez & Lasker, 2003). Environmental influences on larval settlement, such as high sedimentation rates at Sampela or competition and high water flow at Ridge 1, may lead to developmental adaptational responses. Moreover, I. hippuris colonies survive and replicate through external brooding and asexual fragmentation with a propensity for philopatry and upward growth (Dauget, 1992), increasing population structure and expansion on such degraded reefs over time, irrespective of the close proximity of adjacent reefs (ca. 5 km). The population stability of I. hippuris morphotypes through its dual reproductive strategy—if repeated in other locations—has, therefore, the potential to become a reliable indicator of human impacts on coral reefs.

Isis hippuris genetic variability

Phenotypic divergence and biological success of I. hippuris within the WMNP may be a consequence of intraspecific polymorphism due to a high capacity for plasticity with no barriers to gene flow between morphotypes. This, in part, can be a consequence of epigenetic effects, which may be heritable and become fixed through genetic assimilation if conditions persist. This, particularly in the presence of evolutionary capacitance whereby cryptic variation becomes functionally overwhelmed or initiated by the environment, can exert pleiotropic effects on significant developmental processes (Rice, 2008). Such non-additive genetic covariance wields a stronger influence on mutation than random drift (Wagner, Booth & Bagheri, 1997; Rice, 2000; Rice, 2008), itself much stronger in small populations typical of brooding and asexual taxa. Thus, phenotypic divergence as seen in I. hippuris across sites within the WMNP, may be a consequence of hidden genetic variation leading to emergent environmentally mediated fixation accelerated by anthropogenic impact. Peripheral haplotypes reveal emergent lineages (Forsman, 2003), with those at Sampela differing by up to seven base pairs between primary ITS2 sequence comparisons, none of which are shared with other haplotypes within the region. Furthermore, shared haplotypes and thus gene flow at the remaining test sites, (with the exception of haplotype D (sea grass)), suggests assortative mating with the onset of reproductive isolation at Sampela. Thus the more frequent and broadly adapted remaining haplotypes are likely ancestral. Greater sampling with genetic and coalescent analyses is required to confirm such supposition, particularly considering a minority presence of opposing morphotypes at each site (see Rowley, 2014).

In this study the ITS2 region was selected for its ability to detect fine-scale sequence differences (e.g. Coleman & van Oppen, 2008), whereas the highly conserved mitochondrial DNA found throughout the Anthozoa renders nucleotide substitution rates too slow to distinguish at the species or sub-species level (Shearer et al., 2002). Given the renowned caveats associated with the ITS2 region such as intragenomic variation, the secondary structure of each haplotype confirmed molecular morphometric differences. The consistent mutational differences both within (clones) and between sequenced samples, renders PCR or base-calling errors unlikely. The most notable difference was between Sampela (haplotype A) and the remaining sites, yet strong sequence similarities were present between the remaining haplotypes. Furthermore, lack of network reticulation suggests no indication of hybridization, validating confidence in two species taxonomic assignment, emergent or previously diverged. Yet, hybridization at this juncture cannot be overlooked. I. hippuris morphotypes across its distributional range may also represent an Indo–Pacific syngameon as seen in the notoriously diverse and polymorphic scleractinian Acropora Oken, 1815 (Ladner & Palumbi, 2012), with widespread gene flow through introgression (Vollmer & Palumbi, 2002; Ladner & Palumbi, 2012). However, any species delimitation within the Isis genus in addition to I. hippuris is necessary before further inference can be made.

It is clear that pertinent overlap exists between previously described Isis taxa and those present within the WMNP. It is tempting to conclude that I. hippuris is a single species with an extensive phenotypic and geographical range, or that only I. hippuris are present in the Wakatobi with other taxa within the genus elsewhere. Environmentally tolerant taxa tend to possess wide geographic distributions compared to those that are not (Calosi et al., 2010). However, the historically perceived panmixia of I. hippuris is likely more than a single species and not that of a complex when considering similar repetitive phenotypic trait differences across its distributional range. Previous alternative taxonomic assignments are therefore questionable. The standard error of phenotypic variance would be greatly improved by assessing differences between I. hippuris morphotypes with increased specimen analyses from throughout its distributional range; a beneficial strategy when dealing with highly polymorphic taxa. Again, tests of coalescence on numerous independent highly polymorphic markers (SNPs; Ladner & Palumbi, 2012) would be required in order to fully elucidate convergent genotypic-by-environment effects in I. hippuris across its distributional range.

Isididae polyphyly

Phylogenetic analyses confirm haplotype differences as well as polyphyly within the Isididae; a phenomenon recently reported using the putative octocoral mitochondrial marker msh1 (Watling et al., 2012). Even as far back as the earliest part of the last century, Kükenthal (1919) considered the Isididae to be polyphyletic, the subfamilies within as independent groups and the colony axis a “convergent phenomenon.” Furthermore, I. hippuris, itself the type species of this family and the subfamily Isidinae, appears to have minimal phenotypic similarities to virtually all other isidid taxa with the exception of the axis, yet even this has been shown to be scleritic (consist of fused sclerites; Milne-Edwards & Haime, 1857; Kükenthal, 1919; Kükenthal, 1924; Bayer, 1955; Watling et al., 2012; but see Nutting, 1910). Such evidence naturally brings into question the validity of I. hippuris in its current classification. Polyphyly within gorgonian groups across bathymetry is not unknown (McFadden et al., 2006). I. hippuris is the only shallow and zooxanthellate representative of the Isidinae and Isididae respectively, the remainder being characteristic of the deep ocean.

The scleritic composition of the I. hippuris axis further sets it apart from both the Isididae and the suborder Calcaxonians, which are more closely affiliated with the Alcyoniinan-Holaxonian clade as phylogenetically determined by Bernston, Bayer & McArthur (2001) and McFadden et al. (2006). However, this convergent trait holds significant evolutionary intrigue. The fused scleritic internodes with gorgonin nodes of the I. hippuris axis, ensures flexibility and durability under high water energy conditions. Yet what is the selective advantage of a jointed axis in deep-sea isidids? Empirically, this is undetermined but it is not unreasonable to propose that the jointed axis is a relictual anachronism consequential of geological (e.g., opening of the Drake Passage) as well as later glacio-eustatic sea-level changes resulting in bathymetric refugia from turbulent shallow coastal waters (Helm & Schülke, 2003). Thus, the functional significance of an articulated axis at depth is still a mystery; however longer internodes in the colonies at Sampela—like those seen in the benign deep ocean isidids—compared to Ridge 1 were observed but not quantified (SJ Rowley, pers. obs., 2010). Interestingly, deep-sea low flow specialists Isidella Gray, 1857, have long elegant calcareous internodes compared to the larger much more robust internodes of Keratoisis Wright, 1869, characteristic of moderate flow environments in the deep-sea (e.g., 12.75 cm/s, Mohn et al., 2014), yet with no appreciable flexibility (see Rowley, 2014). A deep divergence with stabilizing selection regards a non-sclerite axis in deep-sea isidids may have occurred. Whether the I. hippuris axis is a consequence of convergent evolution based on ecological necessity in heterogeneous environments typical of shallow reefs or deep inheritance is unclear and under investigation.

Conclusion

In closing, the two distinct I. hippuris morphotypes within the WMNP are phenotypically segregated through trait integration between healthy and degraded reefs, likely reinforced through reproductive strategy. The co-variability of light, sediment and water flow between sites fortify directional trait selection (Feder, 1998); colony, branching dynamics, polyp density, sclerite size and diversity all vary significantly between sites. Moreover, polyp and nutrient canals appear canalized due to the additive effect of modules to the colony as opposed to an increase in size. Greater polyp density may lead to an increase in photosynthetic yield and heterotrophy, in turn mitigating and capitalizing on environmental conditions, particularly at Sampela. Diverse phenotypic trait assessment through character trait integration using reciprocal transplant experiments across the two sites would undoubtedly be insightful, particularly as shifts in metabolic function are subject to selection at opposite ends of environmental gradients (Feder, 1998). Selection acts on phenotypic variation (reflecting variation in gene expression), which may have become fixed over time leading to ecological divergence. Isis hippuris morphotypes, tentatively confirmed by ITS2 sequences and secondary structure analyses, have only partial adherence to previously described taxa. Species assignments cannot be prudently made at this time, requiring classical and genomic taxonomic analyses across the distributional range. Furthermore, compelling phylogenetic evidence not only confirms I. hippuris morphotype differences, but also reveals its disassociation within the Isididae. Phylogenetic discernment investigating congruence between skeletal structure, multi-locus next-generation sequencing and coalescence modelling (Puritz et al., 2012), will assist unresolved hypotheses within this turbulent group.

Supplemental Information

Supplemental Information 1 Supplementary Material

Supplementary Material.

Click here for additional data file.

Gratitude is extended to the Wakatobi Government, the Indonesian staff at ALAM of the Wakatobi Marine National Park, and the Wallacea Foundation for their logistical support, particularly Pak Iwan, Arif and Azrul. Special thanks also to Prof RD Gates for lab support and stimulating discussion, as well as Dr S Cairns (USNM), Dr L Van Ofwegen (NCB), and Mr A Cabrinovic (BNHM) for taxonomic support and museum collection access. In addition, many thanks to Dr Z Forsman, J Copus, and L Dueñas for phylogenetic, Arlequin and ITS2 secondary structure analyses guidance respectively. This manuscript benefited from spirited discussion and comments from Dr RL Pyle, Profs Steven M Stanley, SK Davy, J Gardner, and Dr GC Williams, as well as greatly enhanced through the thoughtful reviews from Drs. N Bax and M Taylor.

Additional Information and Declarations

Competing Interests

Author Contributions

Field Study Permissions

DNA Deposition

The authors declare there are no competing interests. Xavier Pochon is an employee of the Cawthron Institute.

Sonia J. Rowley conceived and designed the experiments, performed the experiments, analyzed the data, contributed reagents/materials/analysis tools, wrote the paper, prepared figures and/or tables, reviewed drafts of the paper.

Xavier Pochon conceived and designed the experiments, contributed reagents/materials/analysis tools, reviewed drafts of the paper.

Les Watling contributed reagents/materials/analysis tools, reviewed drafts of the paper.

The following information was supplied relating to field study approvals (i.e., approving body and any reference numbers):

The State Ministry of Research and Technology (RISTEK: No. Surat. Izin: 014/SIP/FRP/SM/VI/2010) granted research permits to Prof DJ Smith, under whose auspices this work was conducted.

The following information was supplied regarding the deposition of DNA sequences:

GenBank accessions KP265675–265702.

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
