# Peer review of "Environmental influences on the Indo–Pacific octocoral Isis hippuris Linnaeus 1758 (Alcyonacea: Isididae): genetic fixation or phenotypic plasticity?"

_PeerJ, doi:10.7717/peerj.1128_

## Round 0.1 · original submission · Minor Revisions

The reviewers enjoyed the paper, and had a few comments that would improve the manuscript. Please address those comments for resubmission.

·

Basic reporting

I think the introduction and methods are very clear. There are a few places where I would like to see more references to support statements but nothing major. The hypotheses are well defined.

Experimental design

In most marine experiments it is difficult to collect environmental factors over the long-term. This experimental design is no exception and understandably so. However, it would be good to have more information about the environmental factors and the length of time / frequency with which their collection occurred in this instance. The variables collected are being utilised as examples of the normal conditions under which the corals studied here exist but this is not explicitly explained and perhaps should be, which caveats about extrapolation of environmental factors added of course.

Validity of the findings

I think the findings of this study are sound and well supported. The language used is cautious where necessary and rightly so.

Additional comments

This is a very well written and detailed paper and I enjoyed reading it.

·

Basic reporting

Broadly in terms of discussion these two areas require some mention:
1) Line 516 - 517
The mitochondrial gene msh is discussed briefly. Due to the use of only one gene, ITS-2 in this study - it would be good to include some mention/discussion of why only ITS data was used and how informative/non-informative it may be. There is a vast literature on this subject - so I am not suggesting a change in the papers focus, just justification so as to emphasise that your results are in fact significant and within the limitations of molecular research on corals ITS was a good choice compared to other genes for the questions you ask.
2) Line 558  
You mention that morphotypes, tentatively confirmed by ITS2 sequences and secondary structure analyses, it might be interesting here to include some discussion or link to the difference between phylogenetic relationships and connectivity/or lack of shown in your haplotype networks.
Also the scale you studied, across 5km, and the difference between populations is informative within the limited literature available for fine scale connectivity between coral reefs - and it would be good to explicitly point this out to emphasise the vulnerability of these ecosystems.

In the rest of the paper I have noted the following for your consideration:

Line 41  Reef biodiversity reflects that of its environment, those within the Indo-Pacific Coral Triangle the most diverse of all.
I can see why you want to say something broad like this, but I'm unsure about it as a sentence. Maybe:
Reef biodiversity is extremely high in the Indo-Pacific coral triangle, in some cases the most diverse of all (ref). High coral biodiversity is heavily correlated with environmental cues (ref)....or something to that effect - as you have two parts in this sentence that require some justification.
Generally this section line 41 - 75 could do with a bit of tweaking to best exemplify the your points.
Line 76 onwards to line 155 - the study really comes into it's own, and the justification/study aims etc become very tidy and succinct. Great job and interesting research!
Line 145  Clearly Isis taxonomy is in a state of flux, compromising conservation efforts due to difficulties in.....x,y,z....be more explicit here.
Line 146  species assignment.
In the impacted site is there lower survival? If this is a new species what will that mean - were they there already and have been able to tolerate the impact up till now?
Line 339 - 340 where you begin to talk about sediment load, maybe elaborate what exactly sediment load is in the context of your study.
Line 443  flow (West et al., 1993; West, 1997; Kim et al., 2004). Rather than these references stating that flow is important, I would like to see some discussion to compare the difference in flow to give some idea of an amount that is good/bad?
Line 469  where you talk about: increasing population structure and expansion on such degraded reefs over time. It's very good that you have this information, and I would like to see some discussion of dispersal in this context. In the context of coral literature, it seems like for your species you have a good idea of reproduction and it would be good to state how this knowledge will help us understand populations dynamics in the context of your study and others.
Line 494  Furthermore, lack of network reticulation suggests no indication of hybridization, validating confidence in two species taxonomic assignment. This subject is not my specialisation, but I was wondering in this context, based on sympatry/co-occurance would there be the capacity to hybridise based on proximity?
Line 540  internodes of Keratoisis Wright, 1869, characteristic of moderate flow environments in the deep-sea - what is moderate flow in the deep sea? Sorry, it was one of my supervisors points the other day within my own writing - if you give a measure make sure it's an informative one.
Line 541  yet with no appreciable flexibility. This is interesting, can you add more context for comparison between deep and shallow populations?
Line 534  geological (e.g., opening of the Drake Passage) as well as later glacio-eustatic sea-level changes and line 535  resulting in bathymetric refugia from turbulent shallow coastal waters (Helm & Schülke, 2003). I find this area very interesting, but if feels a bit like you have gone through the paper to really narrow down your points....then detour to briefly mention the broader implications of deep/shallow and the past. Is there a way to put this more in context of your study/region? As it stands you could delete these lines and it would not change the story behind your paper.

Experimental design

Line 318  - 319 Of the 57 morphological traits measured (59,328 individual measurements), 32 were selected for further analyses (41,265 individual measurements, S2). These numbers are massive in both instances, and I have not used this method, so it's just an idea - but is there some way that the use of so many variables is very statistically informative, but at the same time statistically limiting? If so it might be good to have some mention of this to further justify why these measurements were the final data set.
Line 347 Island with overall haplotype (hd) and nucleotide diversity (π) measured as 0.780 and 0.0197. These measure of nucleotide diversity might require some discussion on comparisons within the literature. Miller et al., 2010 reported nucleotide diversity in 16S and I think in ITS - and there are likely references in the octocoral literature...it would just be informative to see if how your results compare. I had similar numbers for ITS, but I think there might be a difference between deep/shallow estimates? I'm also interested in the amount of variation in your sequences in this context as ITS is traditionally hyper-variable, do you report sequence variation somewhere? I may have missed it, but if not it would be good to have a reference to it. Including some comparison of this would also help to substantiate your claim in line 358  - nucleotide diversity observed across sites and also increase taxonomic certainty. Generally for lines 356 - 358, it would help to be more explicit in your limitations, but also to substantiate that your results are in fact significant within the context of the available literature and current limitations to this type of research.
Line 488  where you mention the presence of opposing morphotypes at each site (SJ Rowley, pers obs). It's good that you have some idea of this, is there any possiblility of discussion on % presence/absence? Or some potential for it in future publications? is pers.obs unpublished data?

Validity of the findings

No comments

Additional comments

Dear Sonia, Xavier and Les,

Thank you for the opportunity to review your paper, I enjoyed learning about your research and look forward to the final publication. There were a couple of areas where I would have liked to see more explanation and coverage. Some of these points, are to be taken into consideration at your discretion, as many of these points are projected from my own personal research interest and experience - namely the fine scale connectivity and conservation aspects, being that this work was carried out in a marine park, there are consequences to this research that were either not discussed, or only discussed in minor detail (potentially due to the format of the paper?).

I wish you the best of luck with your research and publication.

---

## Round 0.2 · accepted · Accept

I would still like you to make a couple of editorial changes before publication.

Please check and correct sentence structure for:

(a) First sentence of introduction, starting 'Reef biodiversity reflects that of its env and geo history, with those within the I-P Coral Triangle being some.....'
(b) Please revise and simplify sentence in introduction starting 'Such diversification may occur.....'
(c) Page 4 line 8: check and revise sentence starting 'In the second scenario....'

Lastly, I am unsure of PeerJ's policy on using unpublished manuscripts in the reference list. I am checking with them so we can be sure. [Note from staff: This is acceptable. Our Production staff will liaise with you as to how to reference them]